# Omphalocele and Associated Anomalies: Exploring Pulmonary Development and Genetic Correlations—A Literature Review

**DOI:** 10.3390/diagnostics15060675

**Published:** 2025-03-10

**Authors:** Dina Al Namat, Romulus Adrian Roșca, Razan Al Namat, Elena Hanganu, Andrei Ivan, Delia Hînganu, Ancuța Lupu, Marius Valeriu Hînganu

**Affiliations:** 1Faculty of Medicine, University of Medicine and Pharmacy “Grigore T. Popa”, 700115 Iasi, Romania; dina.rosca-al.namat@umfiasi.ro (D.A.N.); adrianrosca82@yahoo.com (R.A.R.); dr.elenahanganu@gmail.com (E.H.); ivan.andrei89@gmail.com (A.I.); hinganu.delia@umfiasi.ro (D.H.); marius.hinganu@umfiasi.ro (M.V.H.); 2Department of Surgery II-Pediatric Surgery, 700309 Iasi, Romania; 3“Saint Mary” Emergency Children Hospital, 700309 Iasi, Romania; 4Department of Mother and Child Medicine, University of Medicine and Pharmacy “Grigore T. Popa”, 700115 Iasi, Romania; anca_ign@yahoo.com

**Keywords:** omphalocele, pulmonary abnormalities, anterior abdominal wall, anatomy, exomphalos, pulmonary hypoplasia

## Abstract

Omphalocele is a rare congenital abdominal wall defect, occurring in approximately 3.38 per 10,000 pregnancies. It is characterized by the herniation of abdominal organs through the base of the umbilical cord, enclosed by a peritoneal sac. While omphalocele can occur as an isolated anomaly, it is more commonly associated with congenital syndromes and structural abnormalities. Among its most significant complications, pulmonary hypoplasia (PH) and pulmonary hypertension (PPH) have been shown to negatively impact neonatal prognosis. These conditions result from impaired pulmonary vascular development, leading to respiratory distress and hypoxemia. Unlike many congenital disorders, there is no universally accepted surgical approach for omphalocele repair. The choice of surgical strategy depends on multiple factors, including the size of the abdominal wall defect, presence of herniated solid organs, associated anomalies, and severity of pulmonary complications. Notably, giant omphaloceles are frequently linked to lung hypoplasia, as reduced intra-abdominal space restricts fetal lung expansion, leading to structural lung abnormalities and increased pulmonary vascular resistance. These factors contribute to a higher risk of respiratory morbidity and mortality in affected neonates. This literature review examines the prevalence, significance, and clinical implications of the association between omphalocele and pulmonary abnormalities. Through a systematic analysis of published studies, we evaluated 157 full-text articles along with available titles and abstracts. Our findings indicate that infants with omphalocele often exhibit respiratory complications detectable prenatally and at birth. Severe respiratory insufficiency, particularly due to pulmonary hypoplasia and pulmonary hypertension, significantly increases neonatal morbidity and mortality. While surgical correction may initially exacerbate respiratory challenges, most patients demonstrate short-term recovery with appropriate multidisciplinary management. This review highlights the importance of early diagnosis, comprehensive prenatal assessment, and tailored postnatal management to improve outcomes in newborns with omphalocele and associated pulmonary complications. Further research is needed to establish standardized treatment protocols and optimize long-term respiratory outcomes in these patients.

## 1. Introduction

Omphalocele, also known as exomphalos, is a congenital abdominal wall defect characterized by the herniation of abdominal organs through a midline opening at the base of the umbilical cord. These organs remain enclosed within a three-layered protective membrane composed of peritoneum, Wharton’s jelly, and amnion. The condition occurs in approximately 1 in 1100 pregnancies, but due to a high rate of spontaneous abortions, its incidence in live births is estimated between 1 in 4000 and 1 in 6000 [1]. Based on the extent of the defect and organ involvement, omphalocele can be classified into small, giant, and ruptured types [2]. Omphalocele is often associated with genetic abnormalities, with over 50% of affected infants presenting chromosomal anomalies such as trisomy 13, 18, and 21, as well as syndromic conditions like Beckwith–Wiedemann syndrome [3]. In addition to gastrointestinal complications, up to 24% of newborns with omphalocele exhibit extra-abdominal abnormalities, including cardiac defects and pulmonary hypoplasia [4].

In recent years, studies have highlighted the link between giant omphalocele and abnormalities in both pulmonary parenchyma and vasculature, leading to pulmonary hypoplasia and pulmonary hypertension (PH) [5,6]. The altered development of the thoracic cage further contributes to respiratory complications [7]. Several case reports and series have documented chronic pulmonary issues, including PH, in newborns with omphalocele [8,9,10]. The underlying causes of pulmonary hypertension in these infants are complex.

Pulmonary hypoplasia remains the most frequently reported cause, as demonstrated in post-mortem findings of 14 out of 27 infants with giant omphalocele [9]. Additionally, alveolar capillary dysplasia has been identified as a potential contributor in some cases. Other factors linked to pulmonary hypertension include structural anomalies of the pulmonary arteries, left-to-right cardiac shunts, and congenital heart disease [11,12]. It has been suggested that abnormal prenatal lung development in these infants results from an in utero deformation sequence, influenced by multiple factors [13,14,15]. The displacement of the liver affects thoracic cavity formation, leading to reduced intra-abdominal pressure. Consequently, the lower rib cage flattens inward during fetal breathing movements, further restricting lung development [14]. In many cases, the rectus abdominis muscles attach laterally rather than at the midline, exerting a downward force on the rib cage and contributing to a narrow thorax with slanted ribs [15,16].

Prenatal detection of pulmonary hypoplasia in fetuses with abdominal wall defects has become increasingly feasible through ultrasonographic and MRI assessments. Two-dimensional ultrasound (2D-US) can predict pulmonary hypoplasia by measuring the lung-to-thorax ratio, while the chest-to-trunk length ratio helps identify thoracic restriction [15,16,17,18,19]. More recently, fetal MRI imaging has been used to quantify total lung volumes, offering a more accurate assessment of lung development in fetuses with giant omphalocele [20].

## 2. Materials and Methods

### Electronic Databases and Search Strategy

This review was conducted in accordance with the Preferred Reporting Items for Systematic Reviews and Meta-Analysis (PRISMA) Statement (available at http://www.prisma-statement.org/, accessed on 12 January 2024). PRISMA guidelines were followed throughout the review process to ensure transparency, rigor, and reproducibility in our methodology. The approach involved independent data extraction and quality assessment, which were performed by four researchers, each contributing to the accuracy and thoroughness of the review.

We employed three major electronic databases—PubMed, Web of Science, and EMBASE—to conduct an extensive literature search. These databases were chosen due to their broad coverage of medical and clinical studies, as well as their inclusion of both peer-reviewed articles and the gray literature, ensuring that the search would provide a comprehensive view of the topic. The search was focused on identifying studies that explored the association between omphalocele and pulmonary malformations or pulmonary hypoplasia, specifically examining any correlations between these conditions and their potential effects on lung function and overall respiratory health.

In order to ensure the inclusion of the most relevant and high-quality evidence, we defined a set of inclusion criteria. Initially, we focused on original studies published in English to minimize language bias and ensure accessibility to the broader scientific community. The review included systematic reviews, randomized controlled trials (RCTs), and observational studies, as well as case studies and case reports. These types of studies were selected because they provide a broad spectrum of data—ranging from large-scale quantitative analyses to detailed reports of individual patient experiences—allowing for a comprehensive exploration of the relationship between omphalocele and pulmonary abnormalities.

The time frame for the search spanned from the earliest published studies available up to January 2024, to ensure the inclusion of both foundational research and the latest findings in the field. The inclusion of both historical and recent studies allowed for an exploration of how the understanding of omphalocele and its associated pulmonary complications has evolved over time.

The search process involved the use of specific, well-defined keywords and MeSH (Medical Subject Headings) terms, including “omphalocele”, “pulmonary malformations”, “pulmonary hypoplasia”, “congenital anomalies”, and “lung function”. Boolean operators were employed to refine the search results and ensure that only the most relevant articles were retrieved. After the initial search, the retrieved articles were carefully screened for eligibility based on predefined inclusion and exclusion criteria, focusing on studies that specifically addressed the association between omphalocele and pulmonary issues.

Data extraction was performed independently by each of the four researchers, and any discrepancies between their findings were resolved through discussion and consensus. Following data extraction, the quality of the included studies was assessed using established quality assessment tools appropriate for each study design, ensuring that the final analysis was based on reliable and robust evidence.

By adhering to these procedures, we aimed to provide a comprehensive, objective, and up-to-date synthesis of the literature on the association between omphalocele and pulmonary malformations, contributing valuable insights into this complex area of pediatric medicine.

## 3. Results

The initial PubMed search yielded a total of 6896 articles. To refine the selection, we applied filters to include only studies with available abstracts and free full-text access, reducing the number to 1426. We then used the Medical Subject Headings (MeSH) terms “pulmonary malformation” and “pulmonary hypoplasia” to narrow down the most relevant studies. This search identified 48 and 55 articles, respectively. After a thorough title and abstract screening, 41 studies were determined to be eligible for inclusion in our review. From the Web of Science database, we identified 2339 articles related to omphalocele. To ensure relevance, we conducted a focused search using the terms “omphalocele” AND “pulmonary malformation” and “omphalocele” AND “pulmonary hypoplasia”, which resulted in 110 articles. To further refine our selection, we applied category filters related to Respiratory System, Pediatrics, Obstetrics and Gynecology, Genetics & Heredity, and Surgery, leading to a final dataset of 90 studies for detailed analysis. Our EMBASE search retrieved 4623 articles on omphalocele. When we applied the terms “pulmonary malformation” OR “pulmonary hypoplasia”, we identified 404 and 259 articles, respectively. After filtering for human studies and reviews, 36 and 24 articles met the inclusion criteria. Following duplicate removal and relevance screening, we selected 16 articles from EMBASE for our review. After merging and deduplicating results from all three databases, a total of 876 unique articles were initially considered. Following title and abstract screening, 2199 records underwent further review.

Based on predefined exclusion criteria, 729 studies were removed, including the following: 20 articles filtered out based on Web of Science category restrictions, 603 studies excluded for being non-human research or reviews, 44 duplicates and non-relevant studies removed, and 62 studies excluded after title and abstract assessment. Ultimately, 147 full-text articles were assessed for eligibility. An additional 10 studies were identified through reference screening, leading to a final inclusion of 157 studies in this review.

## 4. Literature Review

During the second step, we revised 147 papers, focusing on the titles, the abstracts, and the full texts in order to select significant data for our review. In total, 157 relevant studies with their citation were included in our reference list. The chest wall is made up of the intercostal muscles, rib cage, and abdominal wall. The only area of the abdominal compartment that can move freely is the ventral abdominal wall since the lateral and posterior abdomen are restricted in their mobility by the back, lower rib cage, pelvis, and iliac crests. The motion of the diaphragm in respect to the thorax provides crucial insight into the mechanics of the chest wall and respiratory pump. Changes in the properties of the abdominal wall influence total respiratory pump operation. Furthermore, congenital anomalies of the ventral abdominal wall may be linked to pulmonary hypoplasia and infant respiratory distress [19,21]. Numerous genes have been associated with the development of the ventral abdominal wall [22,23]. The amount of pulmonary hypoplasia present frequently determines the survival rate of live-born neonates with abdominal wall defects [21].

Abdominal wall abnormalities not only affect respiratory mechanics and diaphragm function postnatally but also impact prenatal lung and chest wall development. Pulmonary hypoplasia in enormous omphalocele may be caused by a variety of factors. Argyle hypothesized that pulmonary hypoplasia is the result of thoracic cage movement impairment, which includes the abdominal musculature and diaphragm [24]. A significant incidence of respiratory insufficiency has been reported to be associated with giant omphalocele. Pulmonary hypoplasia, diaphragmatic dysfunction, and an elongated, narrow thorax have been identified as contributing factors to respiratory issues [25].

### 4.1. Embryology

Embryological Theories of Omphalocele Development

The precise embryological mechanisms leading to omphalocele remain uncertain, with multiple hypotheses attempting to explain its etiology [26]. Gross and Blodgett proposed that restricted body cavity expansion between the 8th and 12th weeks of gestation prevents the midgut from returning to the abdominal cavity after physiological herniation, resulting in omphalocele [27]. Margulies, on the other hand, suggested that the defect arises before the third week of gestation, due to either a failure of the mesodermal transverse septum to unite with its amniotic covering or inadequate proliferation of embryonal connective tissue in the transverse septum, both of which are crucial for the development of the supra-umbilical abdominal wall [28]. Gray and Skandalakis proposed that omphalocele results from a developmental arrest during the physiological herniation of intestines into the umbilical coelom, preventing their normal reintegration into the abdomen [29].

Role of the Diaphragm in Ventral Wall Development

Normal diaphragm development begins with the formation of the pleuroperitoneal fold, a lateral structure that connects dorsally to the mesonephric ridge and ventrally to the transverse septum. During early development, lung buds extend into the peritoneal cavity but do not yet reach the pleuroperitoneal canal. Meanwhile, closure of the pleuropericardial canal separates the pleural and pericardial cavities. At this stage, the developing lungs remain in close proximity to the liver on the right and to both the liver and stomach on the left. As the embryo grows, the pleural cavities expand, leading to a gradual reduction in the peritoneal portion of the lungs. By embryonic day (ED) 13, the posthepatic mesenchymal plate (PHMP) begins to form between the transverse septum, liver, and pleuroperitoneal fold. The PHMP rapidly expands in a laterodorsal direction, staying closely associated with the underlying abdominal organs. By ED 13.5, it forms a ridge that partially covers the liver, while the stomach shifts closer to it on the left side. This stage is also marked by the appearance of the phrenic nerve between the PHMP and the transverse septum. By ED 15, the pleural cavities remain small and cubic, positioned dorsally to both the peritoneal and pericardial cavities. At this stage, the transverse septum has formed the pericardial floor, while the pleuroperitoneal canals remain widely open, with the left canal being larger and more horizontally oval and the right smaller and vertically oriented. The PHMP continues its ventral expansion, interacting with adjacent abdominal organs and gradually closing the pleuroperitoneal canals. The right canal seals approximately six hours before the left, and by ED 17, both are fully closed. As this occurs, the pleural cavities expand rapidly, shifting from a dorsal to a lateroventral position around the pericardial cavity.

Molecular and Mechanical Factors in Ventral Body Wall Formation

Transcription Factor AP-2α has been identified as a key regulator of cell migration, differentiation, and apoptosis—processes critical to ventral body wall formation. Disruptions in these pathways may contribute to defects like omphalocele. The link between abdominal wall defects and lung development is complex. Lung growth requires adequate intrathoracic space, and studies on Mek1flox/flox; Mek2−/−; and Dermo1+/Cre mutants suggest that kyphosis and omphalocele can impose physical constraints on lung expansion, compromising respiratory function and neonatal survival [30]. Additionally, normal lung development relies on amniotic fluid inhalation, which expands airways and promotes alveolarization. Fetal breathing movements in the third trimester further stimulate distal airway and capillary growth [31]. Altered amniotic fluid dynamics in utero may lead to pulmonary capillary remodeling, increasing vascular reactivity and contributing to reduced lung volumes [11].

### 4.2. Epidemiology

A study conducted by the National Birth Defects Prevention Network in the United States analyzed 2308 cases of omphalocele recorded across twelve state population-based registries from 1995 to 2005. The findings indicated an increased likelihood of omphalocele occurrence among infants born to mothers aged 35 and older and those younger than 20 [26].

Additionally, data from the New York State Congenital Malformations Registry (1992–1999) revealed that Black infants had a 70% higher risk of being born with an omphalocele compared to White infants [26]. Maternal obesity, particularly a BMI over 30, has also been identified as a contributing risk factor [26].

Trisomy 18 is the most frequently associated chromosomal abnormality, with 80–90% of affected individuals presenting with an omphalocele [26]. Prenatal ultrasound can detect this condition as early as 13 weeks of gestation, often in conjunction with elevated maternal serum alpha-fetoprotein levels [31]. The incidence of omphalocele ranges between 1 in 4000 and 1 in 10,000 live births, with a female-to-male ratio of 1:1.9 [32,33]. More than half of omphalocele cases involve structural or chromosomal anomalies, with the prevalence potentially reaching 80% [34].

### 4.3. Omphalocele Patients with Pulmonary Abnormalities

The abdominal wall plays a decisive role during inspiration, exhalation, and airway clearance through coughing. The abdominal wall’s compliance plays a crucial role in determining the diaphragmatic motion during inspiration [35]. Similar to what occurs after a large meal, a non-compliant abdominal compartment will restrict diaphragmatic descent, and a non-compliant abdominal wall will impede lower rib cage expansion. However, it is important to note that patients with Prune Belly Syndrome may have a highly compliant abdominal wall. This condition can cause changes in the relationship between the rib cage and abdominal wall, as well as diaphragmatic function, which are corrected when the subjects lie in a supine position [36].

Ewig et al. hypothesized that the extremely flexible abdominal wall prevented the lower rib cage from expanding and allowed the diaphragm muscle fibers to shorten excessively by removing the fulcrum impact of the stomach contents on the lower rib cage. These mechanical drawbacks ultimately led to abdominal paradox—the inward migration of the abdominal wall during inspiration—functional diaphragmatic weakening, and the requirement to recruit auxiliary muscles of inspiration [36]. When the identical subjects were examined in a supine position, all of these findings vanished. Additionally, when the diaphragm was subjected to supine gravity, abdominal viscera exerted cephalad pressure, which improved length–tension relationships and increased area of apposition. Primary closure of the defect in infants with gastroschisis or omphalocele may lead to respiratory compromise, particularly if the abdominal compartment has experienced area loss and there is a significant viscera–abdominal disproportion. It has been postulated that the act of relocating organs to the abdominal compartment leads to inflated intra-abdominal pressures, accompanied by cephalic diaphragmatic displacement and motion restriction [37].

#### 4.3.1. Pulmonary Hypoplasia

Griscom and Driscoll examined the radiographs of a number of stillborn babies and babies who passed away soon after birth. They observed that almost all omphalocele fetuses had noticeably smaller chests [38]. Hershenson et al. similarly observed that within a year, newborns with giant omphaloceles gradually reverted to a normal chest structure [25]. To further, Argyle et al. observed that children with omphalocele who had any large abdominal wall abnormalities in addition to narrow thoracic cages were at a higher risk of developing pulmonary hypoplasia and experiencing respiratory distress [24]. Prenatal ultrasound indicators of pulmonary hypoplasia include a reduced lung-to-thorax transverse area ratio and an elevated chest-to-trunk length ratio [19,39].

In another study, total lung volumes (TLVs) in omphalocele were determined using fetal MRI [20,40]. Infants with giant omphaloceles with less than 50% O/E TLV had poorer Apgar scores at birth, they needed more ventilatory assistance, and they had longer hospital admissions than those with more than 50% [20,41].

#### 4.3.2. Pulmonary Hypertension

Omphalocele has also been associated with the development of pulmonary hypertension. It is often diagnosed with an echocardiography, which uses conventional criteria such as higher right ventricular systolic pressures and septal flattening [42,43]. Partridge et al. found that individuals with giant omphalocele often need pulmonary vasodilator treatment, such as nitric oxide or continuous sildenafil [42].

Patients with omphalocele are more likely to have chronic respiratory issues beyond the first prenatal respiratory distress [43] compared to patients without pulmonary hypertension. Multiple studies have reported cases of left lung collapse and/or narrowing of the left mainstem bronchus in patients with giant omphalocele. Lung collapse, which is occasionally found in this group, could be caused by physical deformation of the bronchus due to increased pressure caused by omphalocele reduction [8,44].

#### 4.3.3. Post-Surgical Respiratory Issues

The primary challenge in repairing this anomaly is of respiratory nature, associated with abdominal hypertension. Circulatory disorders may be evidently linked to the reintegration process [45]. The respiratory impact during management is of particular significance due to the frequent association with pulmonary hypoplasia, which may result in respiratory failure [46,47].

Patients with GO have a reduced abdominal cavity. Therefore, the early closure of the abdominal wall can lead to an abrupt surge in intra-abdominal pressure and to respiratory failure because of the reduced lung capacity. There is a special concern in the case of patients with GO, as they also typically manifest pulmonary hypoplasia, which could result in respiratory failure [48,49]. However, the following factors lead to the impossibility of early surgical treatment: the abdominal–visceral disproportion in neonates, the large diameter of the abdominal wall defect, the presence of large liver tissue in the sac, and other organ anomalies that coexist in infants [50,51].

According to a study conducted by Dimitriou et al., newborns with abdominal wall abnormalities showed a temporary reduction in lung compliance following surgical repair; however, by the third postoperative day, this impact had improved. Nakayama et al. investigated the impact of the closure of abdominal wall abnormalities using pulmonary function testing before and after surgery [52,53].

#### 4.3.4. Diaphragmatic Hernia

Anomalies associated with congenital diaphragmatic hernia (CDH) have been reported in 30–40% of patients [54,55]. Omphalocele, also known as exomphalos, is a congenital anomaly characterized by a midline defect in the abdominal wall that allows abdominal organs, mainly the liver and intestines, to herniate, although they remain covered by a protective membrane. The clinical presentation of omphalocele patients frequently includes pulmonary hypoplasia and/or pulmonary hypertension [40,42,56].

The literature on congenital diaphragmatic hernia (CDH) with an accompanying omphalocele is primarily limited to case reports [57,58], with most of the documented cases being of the anterior or antero-lateral type [59,60,61,62]. In isolated CDH, pulmonary hypoplasia and pulmonary hypertension are the leading factors contributing to higher morbidity and mortality [63,64,65,66,67,68,69,70]. Additionally, pulmonary hypoplasia [20,48] and/or pulmonary hypertension have been identified as independent predictors of survival in cases of isolated omphalocele [6,71]—a finding that is quite unexpected.

The coexistence of these two conditions in the developing fetus is likely to result in a more pronounced degree of pulmonary hypoplasia and/or pulmonary hypertension, which in turn may significantly worsen respiratory insufficiency after birth. This explains why many of these newborns experience a rapid decline in their condition immediately postnatally [72].

According to Harmath et al., Sweed, and Puri, there were approximately 3–4% of CDH cases characterized by an associated abdominal wall defect with an omphalocele (4 out of 100 and 4 out of 116 CDH cases, respectively) [73]. The incidence of CDH associated with an omphalocele was 0.077%, as concluded in a 10-year review of autopsy cases conducted by Borys and Taxy [74]. The clinical management challenges are presented by this uncommon combination of CDH and omphalocele due to the scarcity of literature reviews on their perinatal management. It has been reported that the combination of CDH and omphalocele represented a component of syndromes, including Fryns syndrome and pentalogy of Cantrell [75]. The literature has also documented chromosomal abnormalities, including trisomy 13 [76] and trisomy 18 [73].

The timing of surgical treatment is vital in cases involving both CDH and omphalocele [61]. There was a report that documented CDH resulting in severe respiratory distress or circulatory instability following the closure of the abdominal wall defect [65].

## 5. Chromosomal Abnormalities and Syndromes

Isolated omphalocele is an uncommon congenital defect, occurring in approximately 2–3 per 10,000 live births [77]. These abnormalities are relatively rare, and about 56% of affected individuals exhibit chromosomal abnormalities, including trisomies 13, 15, 16, and 18, as well as Beckwith–Wiedemann syndrome [78,79,80]. Symbrachydactyly, a deformity of the right upper limb, was first identified by Blauth and Gekeler in 1973. This condition is considered a transverse deficit caused by bone dysplasia, with severity categorized into four groups [81,82]. The clinical features are typically consistent, and some cases are associated with pectoralis muscle abnormalities, as seen in Poland syndrome [83]. The frequency of skeletal dysplasia is 2.14 per 10,000 births [84].

Other associated phenotypic features include pulmonary hypoplasia, cardiovascular defects, craniofacial dysmorphism (such as trigonocephaly and forehead prominence), midfacial abnormalities (such as anteverted nares and a long philtrum), gonadal dysgenesis with sex reversal, cryptorchidism, hypospadias, and malformed genitalia, among others. However, the exact correlations between genotype and phenotype remain unclear due to significant variation in phenotypic presentation [82].

Chromosomal abnormalities are commonly associated with fetal growth restriction (FGR), a condition that can be observed as early as the first trimester. Advances in three-dimensional (3D) ultrasound have enabled precise measurements of fetal head and trunk volume, offering significant insights into growth patterns in chromosomally abnormal fetuses. A study conducted on fetuses between 11 + 0 and 13 + 6 weeks of gestation revealed that in trisomy 21 and Turner syndrome, the crown–rump length (CRL) appeared normal, but the head and trunk volume were reduced by 10–15%. In contrast, trisomy 18, trisomy 13, and triploidy exhibited a more significant reduction in trunk volume (about 45%), while CRL remained largely unaffected (under 15%). These findings suggest that chromosomal abnormalities result in generalized growth disturbances, with organ development being more severely affected than skeletal growth [85].

The ability to evaluate the head-to-trunk volume ratio has revealed notable patterns in early-onset FGR for chromosomally abnormal fetuses. The study confirmed that triploidy and trisomies 18 and 13, which are associated with high rates of intrauterine lethality, typically present with severe, early-onset asymmetrical FGR. In contrast, trisomy 21 and Turner syndrome, which have better survival outcomes, display milder, symmetrical growth restriction. These differences in growth patterns are likely due to the variation in fetal and placental development, with different degrees of impairment in both systems [85].

When considering omphalocele in the context of chromosomal abnormalities, particularly in trisomy 13 and trisomy 18, the growth disturbances associated with these conditions have a direct impact on organ development, including the lungs. In these conditions, impaired development of the abdominal wall and lungs can lead to significant pulmonary hypoplasia, making it difficult for the lungs to develop properly. This underscores the importance of early-onset FGR and the interconnectedness of fetal growth and organ development in chromosomally abnormal fetuses. The severe early-onset FGR observed in trisomy 13 and 18 likely exacerbates pulmonary underdevelopment, contributing to the high mortality rates seen in these conditions. Furthermore, the abnormal placental development in these fetuses could also play a role in the reduced fetal circulation and impaired oxygenation, further complicating the growth of the abdominal organs and lungs.

### 5.1. PAGOD Syndrome

PAGOD syndrome is a rare disorder of unknown cause that presents with multiple congenital anomalies and a poor prognosis. It is defined by a combination of malformations—including pulmonary hypoplasia, agonadism (sex reversal), omphalocele, and a diaphragmatic defect—that reflects abnormalities in the pulmonary artery and lung [86]. In most cases, the pulmonary hypoplasia in PAGOD syndrome manifests as a hypoplastic right lung accompanied by a small right pulmonary artery. Additionally, muscular defects in the right hemidiaphragm are frequently observed along with hypogenetic right lung abnormalities [87].

### 5.2. Edward’s Syndrome

Edwards syndrome, also known as trisomy 18, is a chromosomal disorder marked by a wide spectrum of clinical manifestations, multiple congenital malformations, and a high mortality rate. It occurs in approximately 1 in 3000–8000 live births [88,89]. Characteristic features include low birth weight; craniofacial dysmorphism such as microcephaly, micrognathia, and low-set ears; as well as skeletal, renal, and cardiac anomalies. Studies have shown that females with trisomy 18 tend to have a better survival rate compared to males, and survival is also relatively higher among Black infants. The syndrome presents with a variety of phenotypic abnormalities affecting the nervous system, growth, and the structures of the cranium, face, thorax, abdomen, limbs, genitalia, skin, its appendages, and internal organs [90,91].

An abdominal wall defect is observed in fewer than 10% of patients with Edwards syndrome [86,90]. Only about 30% of affected infants survive the neonatal period, and merely 5–10% reach their first birthday [92,93]. Acute cardiopulmonary failure—primarily due to associated cardiac malformations present in 70–100% of cases—is the most common cause of death [94]. 

Omphalocele is frequently observed in fetuses with trisomy 18. A study by Snijders et al. [95] reported that 31% of fetuses diagnosed with trisomy 18 (n = 137) also had omphalocele. Similarly, research by Chen [77] found that 24.1% of fetuses with prenatally detected omphalocele (277 out of 1,148 cases) were affected by trisomy 18. In an analysis of first-trimester ultrasound findings in trisomy 18 cases, Sepulveda et al. [96] identified several common structural anomalies. Among these, 21% of cases presented with omphalocele, while other abnormalities included abnormal hand positioning (6%), megacystis (4%), and anomalies in the four-chamber heart view (4%).

### 5.3. Patau Syndrome (Trisomy 13)

In the United States, Patau syndrome affects approximately 1 in every 12,000 births, with a survival rate of only 10% and a median survival of 7–10 days [77,83,96,97,98,99]. The condition is generally fatal within the first year of life. Key features of Patau syndrome include severe intellectual disability, microphthalmia, cutis aplasia, polycystic kidney disease, holoprosencephaly, cleft lip and palate, low-set ears, polydactyly, congenital heart defects, rocker-bottom feet, and omphalocele [100]. In addition, the disorder impacts multiple systems, including the genitourinary tract, digestive tract, pancreas, liver, kidneys, and lungs [101]. Due to facial deformities, affected infants frequently require postnatal respiratory support—such as oxygenation and ventilation—with many necessitating intubation or tracheostomy [102]. Ultrasound performed after 17 weeks of gestation is the most effective way to identify abnormalities associated with Patau syndrome [103]. Among chromosomal disorders linked to omphalocele, the most frequently observed are trisomy 18 and trisomy 13 [95]. The association between omphalocele and Patau syndrome is particularly significant due to its implications for prenatal diagnosis, genetic counseling, and clinical management. 

### 5.4. Pentalogy of Cantrell

A group of anomalies involving the midline abdominal wall, lower sternum, anterior diaphragm, diaphragmatic pericardium, and an intracardiac defect is termed the Pentalogy of Cantrell [104]. Coleman et al. evaluated a range of disorders arising from the faulty closure of the lateral and craniocaudal embryonic folds, which encompasses conditions such as the Pentalogy of Cantrell, OEIS (omphalocele, exstrophy, imperforate anus, and spina bifida), and LBWC (limb–body wall complex). Their case series demonstrated that the severity of pulmonary hypoplasia proved to be a more reliable prognostic factor than any specific classification system, largely due to the extensive overlap in clinical features among these syndromes [105]. In line with Cantrell’s original observations, common cardiac abnormalities include a ventricular septal defect (100%), atrial septal defect (53%), left ventricular diverticulum (20%), pulmonary stenosis or atresia (33%), and tetralogy of Fallot (20%) [106].

### 5.5. Prune Belly Syndrome (Eagle–Barrett Syndrome)

In 1895, Osler coined the term “Prune Belly Syndrome (PBS)”, even though Frolich had earlier reported a case of congenital absence of abdominal wall musculature in 1839. The condition, also known as triad syndrome or abdominal musculature deficiency syndrome [107], is a rare congenital anomaly occurring at rates of 3.8 per 100,000 live births in males and 1.1 per 100,000 in females [108,109]. PBS is classically defined by a triad consisting of urinary tract malformations, bilateral cryptorchidism, and absence of the anterior abdominal wall muscles, although the specific abnormalities in the abdominal wall and urinary tract can vary in affected females [110].

The characteristic prune-like, wrinkled appearance of the abdominal wall results from either a deficiency or complete absence of its muscular components [111]. In a rare case reported by Gyawali S. et al., an infant presented with a distended abdomen and an omphalocele due to an abdominal wall muscle defect; a mass was noted protruding through the umbilical region. This infant also exhibited bilateral cryptorchidism, but his vital signs and other clinical findings were normal, and no significant facial deformities were observed. Initial laboratory tests showed mild hyponatremia, mild hyperkalemia, and an elevated C-reactive protein level (see Table 1). Blood cultures isolated Enterococcus species, prompting treatment with intravenous Amikacin and Ampicillin based on sensitivity profiles. Additionally, abdominal and pelvic ultrasonography revealed congenital absence of the left kidney. Echocardiography demonstrated mild anterior tricuspid leaflet prolapse with moderate tricuspid regurgitation (gradient of 22 mm Hg), a patent foramen ovale with a left-to-right shunt, and normal biventricular function with a left ventricular ejection fraction of 69% [112].

## 6. Genetic Features

Omphalocele is linked to an increased rate of aneuploidy [113], with trisomy 18 being the most frequently observed chromosomal abnormality [114]. This indicates that when omphalocele occurs alongside other systemic malformations, chromosomal anomalies are significantly more common. Therefore, a prenatal ultrasound that detects an omphalocele should prompt comprehensive structural screening and genetic testing to identify any associated chromosomal issues [115]. Chromosomal microarray analysis (CMA) offers a genome-wide approach capable of detecting imbalanced rearrangements, including small deletions and duplications [116].

Following CMA, whole exome sequencing (WES) can further improve diagnostic yield by approximately 8–10% in fetuses with structural defects [117]. For instance, in one proband, WES identified a heterozygous, potentially pathogenic mutation inherited from the father; such mutations in the COL2A1 gene are frequently associated with various skeletal abnormalities, including dwarfism [118]. Additionally, WES detected a mutation in the SDHB gene inherited from the mother, which is associated with hereditary paraganglioma–pheochromocytoma syndromes (PGL/PCC syndromes, OMIM 115310). Although omphalocele is not commonly linked to SDHB mutations in the literature, the presence of an omphalocele on prenatal ultrasound suggests that the c.725G>A (p.R242H) mutation might contribute to its development [115].

Given the increased risk of fetal aneuploidy in cases of prenatal omphalocele—especially when other ultrasound abnormalities are present [115]—it is advisable to perform routine karyotyping along with CMA testing. If both the karyotype and CMA results are normal, WES should be considered, and if these tests are inconclusive, further molecular methods may be employed to exclude phenotypes such as Beckwith–Wiedemann syndrome (BWS). When the fetal karyotype is normal and no additional anomalies are identified, fetuses with omphalocele generally have a better survival outlook, and elective termination without clear indications is not recommended [115].

Moreover, if a perinatal evaluation reveals holoprosencephaly (HPE), polydactyly, and omphalocele, the possibility of fetal trisomy 13 should be considered. Quantitative fluorescent PCR (QF-PCR) is effective for rapidly confirming trisomy 13 and determining its paternal origin, particularly when cell culture fails, and it provides crucial information for genetic counseling [119]. Chen reported that trisomy 18 was present in 24.1% (277/1148) of fetuses with prenatally diagnosed omphalocele [83]. Furthermore, Sepulveda et al. found that the most common structural defects observed by ultrasound in trisomy 18 were omphalocele (21%), abnormal hand posture (6%), megacystis (4%), and an abnormal four-chamber cardiac view [104].

## 7. Management

The ultimate goal of surgical intervention for omphalocele is to prevent a dangerous rise in intra-abdominal pressure while achieving complete coverage of the fascial and skin layers. Treatment strategies fall into three categories: (1) immediate (primary) repair; (2) staged repair with delayed primary closure; and (3) delayed repair using the “paint and wait” technique.

During the process of reducing the herniated viscera, it is crucial to continuously monitor intra-abdominal pressure. This vigilance helps avoid the development of abdominal compartment syndrome (ACS) along with its associated complications—such as reduced cardiac output, splanchnic hypoperfusion, lactic acidosis, renal failure, intestinal ischemia, and hypoventilation—which can be fatal [120]. Historically, non-surgical management of omphalocele was first described by Ahlfeld in 1899, who used alcohol for its antiseptic and escharotic properties [9]. Later, in 1967, Schuster introduced a staged closure technique employing two Teflon^®^ sheets, a method that paved the way for the “silo” approach. In 1969, Allen and Wrenn refined this technique by using a single, circular Dacron-reinforced silastic sheet attached to the abdominal defect [9].

Although a one-stage, primary closure is ideal because it is associated with favorable survival and morbidity outcomes, it is not always technically feasible—especially in cases where the defect is large and the abdominal domain is insufficient to allow a secure fascial closure [121]. For giant omphaloceles featuring an externalized liver or significant viscero-abdominal disproportion, or in cases accompanied by marked pulmonary hypoplasia and hypertension, the “paint and wait” method is generally preferred [54].

In addition to primary closure, several alternative techniques have been explored over the past three decades, including amnion inversion, the application of prosthetic silos, vacuum-assisted closure, tissue expansion, and the use of various mesh materials, as well as non-surgical escharotic therapy or delayed repair. Despite these advances, no single procedure has emerged as the definitive treatment option [122]. The optimal timing for surgical repair in neonates with giant omphalocele (GO) remains a topic of debate, with limited published guidance available. Delayed closure techniques, such as the Gross procedure—which involves creating skin flaps to cover the sac without incising it—and gradual visceral reduction using a prosthetic silo, have been developed to minimize complications associated with early surgical intervention. Topical treatments like sulfadiazine cream and povidone–iodine, known as the “paint and wait” approach, are also used [123].

For infants with large omphaloceles, a tailored, shared decision-making process with parents is essential. This process should consider factors such as defect size, gestational age, and cardiorespiratory status, as well as the potential need to address other malformations (e.g., diaphragmatic hernia, esophageal or intestinal atresia, congenital heart defects) and appropriate antibiotic prophylaxis. Moreover, the overall care plan should reflect the anticipated trajectory of treatment—such as a palliative approach in cases with trisomy 13 or 18—and take into account the parents’ preferences regarding a more aggressive surgical strategy for earlier closure versus a conservative approach that may involve a prolonged course of dressing changes, frequent outpatient visits, a higher risk of sepsis, and delayed abdominal wall reconstruction [52,124].

The management of GO aims to reduce the herniated contents and close the abdominal defect once the patient is medically stable and adequately supported. However, early closure in patients with small abdominal cavities and trunks can lead to a sudden rise in intra-abdominal pressure and respiratory compromise due to reduced diaphragmatic movement. This concern is particularly significant in patients with GO, who often present with pulmonary hypoplasia and cardiac anomalies that further predispose them to respiratory insufficiency [8,20,124,125,126]. Topical agents, such as povidone–iodine, have shown promise in minimizing pulmonary complications, allowing for hospital discharge under close parental supervision with plans for definitive closure at a later date [126].

The COVID-19 pandemic has adversely impacted many aspects of healthcare, including a decline in the early diagnosis of conditions such as antenatal abdominal wall defects. Social, educational, and financial challenges during the pandemic have also led to delays in seeking timely medical care for the closure of these defects.

## 8. Conclusions

In summary, individuals with omphalocele often exhibit various respiratory complications that can be identified during prenatal assessments and become evident shortly after birth. Severe respiratory failure, particularly due to pulmonary hypoplasia and pulmonary hypertension, substantially increases both morbidity and mortality in these patients [42]. Although surgical repair of the omphalocele may temporarily worsen respiratory function, patients generally experience a short-term recovery.

## Figures and Tables

**Table 1 diagnostics-15-00675-t001:** Congenital syndromes linked to omphalocele and their effects on pulmonary function.

Congenital Anomalies	Association with Omphalocele	Impact on Pulmonary Function
Pentalogy of Cantrell	Multiple congenital defects, including omphalocele, diaphragmatic hernia, ectopia cordis, and sternum defects.	Diaphragmatic hernia can severely impede lung development (pulmonary hypoplasia), causing respiratory issues. Ectopia cordis and chest wall defects might impair the thoracic cavity, resulting in limited lung expansion.
Patau Syndrome (Trisomy 13)	Often feature omphalocele as part of a spectrum of abnormalities.	Pulmonary hypoplasia that may result in chronic respiratory distress. Associated congenital cardiac abnormalities may further impair pulmonary function.
Prune Belly Syndrome (Eagle–Barrett Syndrome)	Some cases of Prune Belly Syndrome may present omphalocele.	The deficiency of abdominal musculature might hinder efficient breathing mechanics, complicating lung expansion. Pulmonary hypoplasia may also arise from the underdevelopment of the thoracic cavity, especially in cases of oligohydramnios during gestation.
PAGOD Syndrome	Multiple congenital defects, including omphalocele, diaphragmatic defects, pulmonary hypoplasia, gastrointestinal defects, and agonadism.	Pulmonary hypoplasia is a key characteristic of PAGOD. Omphalocele and diaphragmatic anomalies (common in PAGOD)
Edward’s Syndrome (Trisomy 18)	Omphalocele is a common congenital defect in Edward’s syndrome (30–50%).	Mechanical compression following surgical repair. Pulmonary hypoplasia frequently occurs in newborns with omphalocele and Edwards syndrome. Diaphragm displacement. Hypotonia. Congenital heart defects.

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
