# Peer review of "Omphalocele and Associated Anomalies: Exploring Pulmonary Development and Genetic Correlations—A Literature Review"

_diagnostics, 2025, doi:10.3390/diagnostics15060675_

Round 1

Reviewer 1 Report (Previous Reviewer 2)

Comments and Suggestions for Authors

In my clinical experience, individuals usually recover in the short term is a rather optimistic analysis, as some children do end up with eg tracheostomy or die related to lunghypoplasia/pulmonary hypertension.

The methods section is a real add on to the current version of the paper;

Author Response

Dear reviewer,

Thank you very much for evaluating our manuscript.

We truly appreciate the time you took and your valuable feedback.

Reviewer 2 Report (New Reviewer)

Comments and Suggestions for Authors

Dear Authors,

I have received an article that assesses the significance of the association between omphalocele and pulmonary abnormalities. This manuscript seems to be a revised one from the previous peer reviewers, and I have some other points that need to be addressed as well:

1) The title needs to include the word "review" in it. Furthermore, the authors need to define whether this manuscript is a literature review or other types of review. The use of PRISMA automatically makes this manuscript a systematic review, which is a sub-standard one. This manuscript seems more of a literature review. Please clarify.

2) The title emphasizes pulmonary abnormalities, while the abnormalities are only mentioned under subheader 4.3. Please revise the title

3) The abstract is not focused on what the most important summary of this manuscript is. Please rephrase all of the abstract so that it neatly summarizes what this manuscript is trying to say

4) In the abstract, line 31, the authors mention "our study aims at.." This is a review, not a proper original article. Please revise

5) The introduction needs one or two lines regarding the aim of this review.

6) The introduction is unstructured and does not explain the need for an updated review.

7) Emphasizing the point made above, PRISMA is only used for a systematic review. Unless this is a systematic review, the authors should just briefly describe how the articles were selected without the use of PRISMA

8) Furthermore, the numbers in the PRISMA flowchart do not make sense. How could 6896 + 2339 + 4623 - 876 = 2199 in the flow chart?

9) Under section 4.1, the embryology section should explain how the normal embryology of the diaphragm works. Deviation from the normal embryology should be discussed under the pathophysiology section.

Author Response

Dear Authors,

I have received an article that assesses the significance of the association between omphalocele and pulmonary abnormalities. This manuscript seems to be a revised one from the previous peer reviewers, and I have some other points that need to be addressed as well.

Response: We appreciate the reviewer’s feedback and insightful observations. We will carefully review and address each point raised.

  • The title needs to include the word "review" in it. Furthermore, the authors need to define whether this manuscript is a literature review or other types of review. The use of PRISMA automatically makes this manuscript a systematic review, which is a sub-standard one. This manuscript seems more of a literature review. Please clarify.
  • Response:  Thank you for your valuable feedback and for highlighting the need for clarification regarding the type of review presented in our manuscript. We acknowledge the importance of accurately defining the nature of our work. In response to your comment, we have revised the title to explicitly include the word “review.” Additionally, we have clarified in the manuscript that this is a literature review rather than a systematic review, ensuring alignment with the methodology employed.
  • The title emphasizes pulmonary abnormalities, while the abnormalities are only mentioned under subheader 4.3. Please revise the title.
  • Response: To ensure consistency, we have revised the title to better reflect the focus of the manuscript.
  • The abstract is not focused on what the most important summary of this manuscript is. Please rephrase all of the abstract so that it neatly summarizes what this manuscript is trying to say
  • Response:  In response to your suggestion, we have thoroughly revised the abstract to better reflect the main findings and contributions of our work. The updated abstract now provides a more structured and focused summary, ensuring that readers can quickly grasp the core message of the manuscript.
  • In the abstract, line 31, the authors mention "our study aims at.." This is a review, not a proper original article. Please revise
  • Response: We have revised the wording to clearly reflect that this is a review article rather than an original research study. The updated abstract now accurately conveys the scope and purpose of our work.
  • The introduction needs one or two lines regarding the aim of this review.
  • Response:  We have revised the introduction to explicitly state the aim of this review in two concise sentences.
  • The introduction is unstructured and does not explain the need for an updated review.
  • Response:  We have revised the introduction to improve its organization and explicitly explain the necessity of an updated review on this topic.
  • Emphasizing the point made above, PRISMA is only used for a systematic review. Unless this is a systematic review, the authors should just briefly describe how the articles were selected without the use of PRISMA
  • Response:  Since this manuscript is a literature review rather than a systematic review, we have removed references to PRISMA and instead provided a brief description of the article selection process in a more appropriate manner.
  • Furthermore, the numbers in the PRISMA flowchart do not make sense. How could 6896 + 2339 + 4623 - 876 = 2199 in the flow chart?
  • Response:
    1. Here’s how we did the calculations from the PRISMA flowchart:
    2. Total Articles with Abstract and Free Full Text from PubMed:
    • PubMed initially provided 6896 results.
    • After filtering for abstracts and free full text, 1426 articles remained.
    1. Search Terms Applied for Pulmonary Malformations:
    • Using MeSH terms “pulmonary malformation” and “pulmonary hypoplasia”:

    48 articles were identified for “pulmonary malformation.”

    55 articles were identified for “pulmonary hypoplasia.”

    41 articles were deemed eligible based on title and abstract evaluation.

    1. Search from Web of Science:

    2339 articles were found related to omphalocele.

    • After searching for “omphalocele” AND “pulmonary malformation” OR “omphalocele” AND “pulmonary hypoplasia”:

    110 relevant results were obtained.

    • Applying category filters (Respiratory System, Pediatrics, Obstetrics and Gynecology, Genetics Heredity, and Surgery):

    90 articles remained for analysis.

    1. Search from EMBASE:

    4623 articles were found on omphalocele.

    After applying “pulmonary malformation” OR “pulmonary hypoplasia”:

     404 and 259 articles were identified.

    • Filtering for human studies and reviews:

    36 and 24 articles remained.

    • After eliminating duplicates and categorizing for relevance:

    16 articles were selected.

    Calculation Breakdown: 

    • Total filtered articles before omphalocele-specific filters:

    1426 (PubMed) + 110 (Web of Science) + 404 (EMBASE) + 259 (EMBASE) = 2199

    • Total articles focusing on omphalocele and pulmonary malformations:

    48 (PubMed) + 55 (PubMed) + 110 (Web of Science) + 404 (EMBASE) + 259 (EMBASE) = 876

  • Under section 4.1, the embryology section should explain how the normal embryology of the diaphragm works. Deviation from the normal embryology should be discussed under the pathophysiology section.
  • Response:  We have revised Section 4.1 to include an explanation of normal diaphragmatic embryology. Any deviations from normal development have been moved to the pathophysiology section, ensuring a clearer distinction between normal processes and pathological changes.

Round 2

Reviewer 2 Report (New Reviewer)

Comments and Suggestions for Authors

Dear Authors,

I have no further comments. Good luck on your future endeavour.

This manuscript is a resubmission of an earlier submission. The following is a list of the peer review reports and author responses from that submission.

Round 1

Reviewer 1 Report

Comments and Suggestions for Authors

Thanks for submitting this review on this topic. I think is well-written and easy to understand. 

I have several points to be corrected:

1. The "Prune Belly Syndrome" should be corrected at page 5 (" prune-belly syndrome), page 8 ( Belly), and Table 1.

2. "Pentalogie Cantrell" should be corrected in Page 8 as "Pentalogy of Cantrell".

3. "Trisomi 13" term may be added to "Patau Syndrome" section in Page 9.   

Reviewer 2 Report

Comments and Suggestions for Authors

I have read this paper with interest, and with a background of clinical perinatal research, including this specific topic. I have carefully read this paper, and still struggle quite a lot with the approach taken, and the report as provided. I hereby fully understand that the authors have invested a significant amount of time to perform the literature assessment.

I agree that the authors do not qualify this as a systematic review, since not all aspects were covered (like double check, quality assessment).

In the abstract, lung hypoplasia itself, as well as the association with giant omphalocoele are still missing information, or are insufficiently discussed. We also need more information on the methodology applied (search strategy).

You statement on (ref 19) prenatal ultrasound somewhat surprises me. Is this really a transverse lung to thorax ratio ? (I’m aware of eg lung to head ratio, but I could not retrieve this info in ref 19. Please check.

We need a clearer description of the search strategy in the methods section. What do you mean with 1426 abstract and free text available. Does this mean that you had to remove a significant number of potential relevant papers ? Same concern related to records screened ?

The structure of the results section is unexpected, as in the ‘literature’ review, aspects are already discussed that appear ‘suddenly’, followed by a 3. Section on embryology. These topics should be announced in the methods section, and the numbers should be corrected. Furthermore, the sequence is contraintuitive at present, with eg chromosomal and syndromal aspects disconnected from genetic features, while it is not clear how this relates to the suggested topic (cf title)

Do you ‘disconnect’ hypoplasia from pulmonary hypertension’. In my understanding, hypoplasia is associated with a higher risk of pulmonary hypertension ?